# Advanced Algorithm for Reliable Quantification of the Geometry and Printability of Printed Patterns

**DOI:** 10.3390/nano13101597

**Published:** 2023-05-10

**Authors:** Jongsu Lee, Chung Hwan Kim

**Affiliations:** 1Department of Advanced Components and Materials Engineering, Sunchon National University, 255 Jungang-ro, Suncheon 57922, Republic of Korea; jslee0505@scnu.ac.kr; 2Department of Mechanical Engineering Education, Chungnam National University, 99 Daehak-ro, Daejeon 34134, Republic of Korea

**Keywords:** printed electronics, printing, geometry, printability, quantification

## Abstract

In nanoparticle-based printed electronic devices, the printability of the patterns constituting the device are crucial factors. Although many studies have investigated the printability of patterns, only a few have analyzed and established international standards for measuring the dimensions and printability of shape patterns. This study introduces an advanced algorithm for accurate measurement of the geometry and printability of shape patterns to establish an international standard for pattern dimensions and printability. The algorithm involves three core concepts: extraction of edges of printed patterns and identification of pixel positions, identification of reference edges via the best-fitting of the shape pattern, and calculation of different pixel positions of edges related to reference edges. This method enables the measurement of the pattern geometry and printability, including edge waviness and widening, while considering all pixels comprising the edges of the patterns. The study results revealed that the rectangle and circle patterns exhibited an average widening of 3.55% and a maximum deviation of 1.58%, based on an average of 1662 data points. This indicates that the algorithm has potential applications in real-time pattern quality evaluation, process optimization using statistical or AI-based methods, and foundation of International Electrotechnical Commission standards for shape patterns.

## 1. Introduction

Printed electronics is considered a promising candidate for mass production of flexible electronic devices owing to its flexibility, environmental friendliness, and cost-effectiveness [1,2,3,4,5,6]. Nanoparticle-based printing and coating methods are essential for the fabrication of flexible electronic devices comprising several patterns, such as lines and shapes, formed from single or multiple layers. The quality and printability of the pattern are crucial factors affecting the outcomes of printed electronics [7,8,9,10].

Thus far, several studies have focused on improving the printability of functional patterns. Kapur analyzed the effects of the ratio of web speed to roll speed, that is, the speed ratio and ink properties on the ink transfer ratio during doctoring and ink transferring processes [11]. Kitsomboonloha et al. comprehensively analyzed the ink transfer ratio in the ink transferring phase of direct gravure printing, that is, ink filling, doctoring, and ink transferring phases, according to the capillary number of ink [12]. In addition, they studied the effects of doctoring conditions, such as the geometry of the doctor blade and blade angle, and the speed parameter on the thickness of the lubrication film in the doctoring phase [13]. The air volume in the cell in the ink filling phase of direct gravure printing was studied by Cen et al. in terms of the capillary number of ink [14]. Raske et al. theoretically analyzed the ink transfer ratio during the ink transferring phase of reverse gravure coating in terms of the speed ratio, capillary number, and web tension [15]. In these studies [11,12,13,14,15], printability was evaluated based on the net ink transfer ratio during the printing and coating. Nguyen et al. theoretically and experimentally studied the effects of the surface tension of ink, surface energy, and the aspect ratio of the cell on the ink transfer ratio and pattern width [16]. Moreover, they studied the relationships among printing conditions such as printing speed, nip pressure, and contact angle of ink on web and pattern width [17]. Hyun et al. studied the printed pattern width in terms of the width of line openings in a screen mask for screen printing [18]. Park et al. experimentally analyzed the widening effect because of the engraving type (groove and dot) and aspect ratio of the engraved patterns [19]. Noh et al. and Park et al. fabricated thin film transistor active matrices using roll-to-roll direct gravure printing. Consequently, they evaluated the printability of the component patterns of the transistors using the line width, edge waviness, and surface roughness [20,21]. Lee et al. defined the criteria for evaluating the printability as the continuity, average width, and local smudging. They statistically optimized the printing conditions using these criteria based on the Box–Behnken design of experiments [22,23,24]. The effects of web speed and ink viscosity on the pattern width were experimentally analyzed by Nam et al. and Lee et al. [25,26]. Further, Jeon et al. evaluated line and mesh patterns using width, edge waviness, and pinhole ratio [27]. Lee et al. analyzed the relationship among the ink viscosity, printing speed, pattern width, and length of the tailing defects affecting the edge waviness [28]. The printability of the pattern was evaluated on the basis of the pattern width and edge waviness in [16,17,18,19,20,21,22,23,24,25,26,27,28].

Previously reported results indicate that printability has been defined by researchers as an ink transfer ratio [11,12,13,14,15], pattern width (or widening ratio) [16,17,18,19], or edge waviness [20,21,22,23,24,25,26,27,28] depending on the perspective; however, printability cannot be defined as a single term. The Printed Electronics Technical Committee of the International Electrotechnical Commission (IEC/TC119) has defined printability from the perspective of international standards, and related standards are being actively established [29].

Considering that printability directly affects the performance and reliability of devices, the appropriate management and control of the printability of patterns during the printing process can enhance device quality and increase productivity. IEC/TC119 has established several international standards for printability, including measurements of the line pattern width, void, and waviness [29,30]. Although shape patterns are essential for printing the component layers of flexible electronic devices, there exists no accepted international standard for the measurement of the dimensions and printability of shape patterns.

Therefore, to establish an international standard for the measurement of the dimensions and printability (widening and edge waviness) of patterns, this study proposed an advanced algorithm that ensures reliable quantification of the geometry and quality of shape patterns that are most frequently used in printed electronic devices using image processing. Moreover, the experimental verifications have shown that the proposed algorithm accurately determines the geometry and quality of the pattern.

## 2. Methods

Figure 1a,b shows the proposed quantification method and its flowchart, respectively. The method has three core ideas: (1) extracting the edges of printed patterns and identifying the positions of pixels constituting the edges; (2) identifying the reference edges (reference sides and circles for rectangular and circular patterns, respectively), which are the best-fitting of the shape pattern; and (3) calculating the different pixel positions of the edges related to the reference sides or circles. Using this method, one can measure the geometry of the pattern and quantify its printability, including widening and edge waviness, considering that all pixels constitute the edges of the patterns.

The process for quantifying the geometry and printability of a pattern is shown in Figure 2 and involves five steps.

**Step 1.** Preparation of the printed pattern to be measured (Figure 2a).

**Step 2.** Capturing an image of the pattern using a microscope or camera. The entire image has patterned and non-patterned areas (Figure 2b). For a rectangular pattern, in cases of an oblique extracted image, it should be rotated such that one of its sides is parallel to the *x*- or *y*-axis using a rotation matrix (Figure 2c); a captured image may be tilted by angle θ owing to the oblique angle of its design or by the rotation of the sample in an optical measurement device. If a design pattern is tilted, the tilt angle of the image is dependent on tilting angle θ of its design pattern. If a sample is positioned on a stage of the device while being rotated by angle θ, the tilt angle of the image can be obtained by determining the angle at which a line parallel to the reference line, which is one of the lines composing the shape, appears following the rotation of the shape.

**Step 3.** The edges of the pattern that distinguish the patterned and non-patterned areas are identified, the coordinates of each pixel that constitutes the edge are obtained, and the pattern image is converted into a brightness matrix using image-processing software such as MATLAB (Figure 2d), and the pattern sides are extracted from the matrix using a Sobel operator, depending on the minimum brightness level of the pattern (Figure 2e).

**Step 4.** The reference edges that are the best-fitting lines for the edges of printed patterns are identified. The reference edges correspond to the four sides in the case of a rectangle, or a circle in the case of a circular pattern. Reference edges can be obtained by applying the least squares method to the coordinates of the pixels on each side.

**Step 5.** Measure the geometry of the pattern and/or evaluate the printability by calculating the edge waviness related to the reference edge and widening of the pattern compared to the dimensions of the original design.

### 2.1. Measurement Method for Rectangle Pattern

Figure 3 shows a schematic for the measurement of the geometry of a rectangle from the extracted pattern edges in Figure 2e. First, vertical and horizontal reference lines that are parallel to the sides of the rectangle are drawn (Figure 3a). The lengths of the horizontal and vertical reference lines extend over the horizontal and vertical sides of the target pattern, respectively. All points comprising the vertical and horizontal sides can be employed as sample points via the calculation of the average distance between the pair of pixels constituting the horizontal (left and right) and vertical (top and bottom) boundaries and their corresponding horizontal and vertical reference lines, respectively. Subsequently, the distance between the pixels of the left *W_L_*(*n*) and right *W_R_*(*n*) sides can be determined on the basis of the vertical reference line. Similarly, the distance between the pixels of the top *H_T_*(*n*) and bottom *H_B_*(*n*) sides can be obtained from the horizontal reference line. The width of the rectangle (*W*) can be determined by calculating the average distance between the pair of pixels constituting the left and right boundaries and their corresponding vertical reference lines. Similarly, the height of the rectangle (*H*) can be determined by calculating the average distance between the pair of pixels that form the top and bottom boundaries, and the corresponding horizontal reference lines. The center of the rectangle (*W_C_*, *H_C_*) can be obtained by averaging *W_L_* and *W_R_*, and *H_B_* and *H_T_*, respectively. These measurements are represented mathematically in Equations (1)–(3) as
(1)$$W=\sum_{n=1}^{N_{W}}\frac{\left(W_{R}(n)-W_{L}(n)\right)}{N_{W}}$$
(2)$$H=\sum_{n=1}^{N_{H}}\frac{\left(H_{T}(n)-H_{B}(n)\right)}{N_{H}}$$
(3)$$\left(W_{C},H_{C}\right)=\left(\frac{1}{N_{W}}\sum_{n=1}^{N_{W}}\frac{\left(W_{R}(n)+W_{L}(n)\right)}{2},\;\frac{1}{N_{H}}\sum_{n=1}^{N_{H}}\frac{\left(H_{T}(n)+H_{B}(n)\right)}{2}\right)$$

Using the above information, the extent of widening (*w*, *h*) can be calculated by comparing the pattern with the reference design in Equations (4) and (5).
(4)$$w=\frac{W-W_{0}}{W_{0}}$$
(5)$$h=\frac{H-H_{0}}{H_{0}}$$
where *w* and *h* are the ratios of widening in the width and height of the printed rectangular pattern, respectively; and *W*_0_ and *H*_0_ are the width and height of the original rectangular design, respectively.

Using Equations (6)–(9), we can obtain the average (*D_mX_* and *D_mY_*) and standard deviations of the distance between the sides of the rectangle and its centerlines (*D_sX_* and *D_sY_*). The reference sides *W_L_*_0_, *W_R_*_0_, *H_B_*_0_, and *H_T_*_0_ were obtained by applying the least-squares method to *W_L_*(*n*), *W_R_*(*n*), *H_T_*(*n*), and *H_B_*(*n*) (Figure 3b). Because the corners of the rectangle have curvatures, the data near the corners should be excluded while applying the least-mean-square. This can be achieved by selecting only those pixels whose distance falls within some boundaries, such as *D_m_* − 3*D_s_* and *D_m_* + 3*D_s_* on each side (Figure 3c).
(6)$$D_{mX}=\sum_{n=1}^{N_{W}}\frac{\left|W_{X}\left(n\right)-W_{C}\right|}{N_{W}}$$
(7)$$D_{mY}=\sum_{n=1}^{N_{H}}\frac{\left|H_{Y}\left(n\right)-H_{C}\right|}{N_{H}}$$
(8)$$D_{sX}=\sqrt{\sum_{n=1}^{N_{W}}\frac{{\left(\left|W_{X}\left(n\right)-W_{C}\right|-D_{mX}\right)}^{2}}{N_{W}}}$$
(9)$$D_{sY}=\sqrt{\sum_{n=1}^{N_{H}}\frac{{\left(\left|H_{Y}\left(n\right)-H_{C}\right|-D_{mY}\right)}^{2}}{N_{H}}}$$
where *X* and *Y* indicate the position of the sides of the rectangle on the x-axis and y-axis, that is, the left (*L*) and right (*R*) sides, and the bottom (*B*) and top (*T*) sides, respectively.

Finally, we obtain the edge waviness for each edge (Figure 3d), given by Equations (10) and (11), and calculate the root-mean-square as the statistical data for edge waviness using Equations (12) and (13).
(10)$$E_{X}(n)=W_{X}(n)-W_{X0}$$
(11)$$E_{Y}(n)=W_{Y}(n)-W_{Y0}$$
(12)$$e_{X}=\sqrt{\sum_{n=1}^{N_{W}}\frac{E_{X}^{2}(n)}{N_{W}}}$$
(13)$$e_{Y}=\sqrt{\sum_{n=1}^{N_{H}}\frac{E_{Y}^{2}(n)}{N_{H}}}$$

### 2.2. Measurement Method for Circle Pattern

Figure 4a shows a schematic of the radius of the circle. It is necessary to determine the center of the circle before measuring its radius. Figure 4b–d illustrates how to identify the center of a circular pattern using lines passing through the edge. Two lines (*L* and *M*) are drawn through any two points on the edge (Figure 4b), while two lines (*l*_1_ and *m*_1_) are drawn perpendicular through the centers of *L* and *M*, respectively (Figure 4c). The intersection of these two lines is the center of the circular pattern (*W_C_* and *H_C_*). If we select a pair of points on the edge for lines *L_n_* and *M_n_*, that is, ((*Lx_n_*_1_, *Ly_n_*_1_) and (*Lx_n_*_2_, *Ly_n_*_2_)), and ((*Mx_n_*_1_, *My_n_*_1_) and (*Mx_n_*_2_, *My_n_*_2_)), respectively, we can obtain the intersection points (*Wc* and *Hc*) using Equations (14) and (15):(14)$$W_{C}=\frac{\left(ly_{mn}-\alpha_{2}lx_{mn}\right)-\left(ly_{ln}-\alpha_{1}lx_{ln}\right)}{\alpha_{1}-\alpha_{2}}$$
(15)$$H_{C}=\alpha_{1}W_{C}+\left(ly_{ln}-\alpha_{1}lx_{ln}\right)$$
where
$$\alpha_{1}=\frac{Lx_{n1}-Lx_{n2}}{Ly_{n2}-Ly_{n1}},\;\;\;\alpha_{2}=\frac{Mx_{n1}-Mx_{n2}}{My_{n2}-My_{n1}}$$
$$lx_{ln}=\frac{Lx_{n1}+Lx_{n2}}{2},\;\;ly_{ln}=\frac{Ly_{n1}+Ly_{n2}}{2},\;\;lx_{mn}=\frac{Mx_{n1}+Mx_{n2}}{2},\;\;ly_{mn}=\frac{My_{n1}+My_{n2}}{2}$$

If the edge of the circle is not uniform, an additional line (*l_n_* and *m_n_*) passing through two points on the edge (*L_n_* and *M_n_*) may result in more than two additional centers (*W_Cn_* and *H_Cn_*). Thus, to obtain the statistically converged coordinates of the center, the average of the multiple sets of center positions can be considered. First, we obtain the median of the *x* and *y* coordinates of the points constituting the target circle (*W_m_*, *H_m_*). Subsequently, the center positions in the range of 0.9–1.1 times the derived medians (*W_C_*, *H_C_*) are extracted. Let *W_C_*_0_ and *H_C_*_0_ be the averages of *W_C_* and *H_C_*, respectively.

The mean (*Rad_m_*) and standard deviation of the circle radius (*Rad_s_*) can be obtained by calculating the distance between the points constituting the edge and center of the circle, as expressed by Equations (16) and (17), respectively.
(16)$$Rad_{m}=\sum_{n=1}^{N}\frac{\sqrt{{\left(W(n)-W_{c0}\right)}^{2}+{\left(H(n)-H_{c0}\right)}^{2}}}{N}$$
(17)$$Rad_{std}=\sqrt{\sum_{n=1}^{N}\frac{{\left(\sqrt{{\left(W(n)-W_{c0}\right)}^{2}+{\left(H(n)-H_{c0}\right)}^{2}}-Rad_{m}\right)}^{2}}{N}}$$
where *W*(*n*) and *H*(*n*) are the *x* and *y* coordinates of the *n*th point, comprising the boun-dary of the printed circular pattern, respectively, and *N* is the number of points constituting the circle.

The radius of the printed circle pattern calculated at each pixel of the edge of the circle as well as the widening of the circle can be obtained as:(18)$$R(n)=\sqrt{{\left(W(n)-W_{c0}\right)}^{2}+{\left(H(n)-H_{c0}\right)}^{2}}$$
(19)$$c=\sum_{n=1}^{N}\frac{R(n)-R_{0}}{N}$$
where *R*_0_ is the radius of the original design of the circle pattern.

## 3. Applications

The proposed method was used to measure the geometry of printed patterns and quantify their printing quality. Milliscale rectangles (width = 2 mm and height = 2 mm) and circular patterns (diameter = 1.7 mm) were printed to experimentally verify the superiority of the proposed algorithm. Patterns were printed using an industrial-scale roll-to-roll machine (AMP Korea Co., Ltd., Gyeonggi-do, Republic of Korea). Figure 5a,b shows the roll-to-roll machine and direct gravure printing process used in this study, respectively. In Figure 5a, the roll-to-roll machine comprises an unwinder, an in-feeder, the first and second printer, the first and second dryer, an out-feeder, and a rewinder; the width of the web is 100 mm. The range of tension and web speed were 1–196 N and 1–20 m/min, respectively. Figure 5c shows the ink transferring procedure of the direct gravure printing. Gravure printing is among the most suitable techniques for the mass production of printed electronic devices. The ink transfer in gravure printing can be classified into four phases: ink filling, doctoring, ink transfer, and ink setting [11,16]. During the ink filling phase, ink is provided to the engraved cell. The fractions of ink that are deposited on the surface of the printing roll are wiped out using a doctor blade in the doctoring phase. During the ink transferring phase, the ink in the engraved cell is transferred to the web by the nip roll. Finally, the ink transferred to the web is widened or agglomerated depending on the surface tension of ink and surface energy of substrate in the ink setting. As mentioned in our previous study, ink transfer in the gravure printing process is very complex, and even minor mismatches in the process conditions, such as the ink, web characteristics, and machine conditions, can significantly affect the quality of microscale patterns. Quantification of the geometry and quality of the printed pattern are essential for determining the process conditions using statistical optimization (e.g., design of experiments) and artificial intelligence (AI)-based techniques [22]. In our previous study, we determined the surface tension and viscosity of the ink as 37 mN/m and 10,000 cP, respectively, and the surface energy of the web as 50 mN/m to minimize the widening of the printed pattern [16,22]. Table 1 lists the properties of the ink, web, and machine used in the experiment. 

Figure 5d,e presents the scanning electron microscopy (SEM) image and the contact angle of the ink solvent under the printing conditions listed in Table 1. The average particle size of the ink was 150 nm and the contact angle was 48.44°.

Figure 6a–f shows: (a) the printed pattern image, (b,c) extracted edges of the pattern, (d) widening, (e,f) histogram showing the waviness of the pattern, (e) and each side (f). In Figure 6b,c,f, the red, green, blue, and cyan bold lines represent the top, bottom, left, and right edges of the pattern, respectively, whereas the dotted and dashed lines in Figure 6b,c represent the designed pattern and reference sides, respectively.

The number of data points used to obtain the widening and edge waviness in the x-axis and y-axis (shown in Figure 6d) was 1968. Thus, the proposed method in Section 2.1 facilitated successful extraction of the edges of the pattern sides. Consequently, the widening and edge waviness of the pattern can be quantified, as shown in Figure 6d–f. The pattern was widened by approximately 0.13 mm (6.5% of the pattern width) in the x-axis, as shown in Figure 6d, owing to tailing [11,18] and the spreading of the pattern at the imperfect dots at the edges of the engraved pattern [23]. Further, the average, standard deviation, and a maximum deviation of the pattern width were calculated to be 0.0105, 0.0084, and 0.0514 mm, respectively, owing to the edge waviness. In Figure 6f, the standard deviations of the top, bottom, left, and right sides are 0.0138, 0.0069, 0.0153, and 0.0159 mm, respectively. Thus, the pattern’s sides were widened and exhibited wide deviations, except for the bottom side. Considering the printing direction in Figure 6a and the results of our previous studies [23,24], the top side as well as the left and right sides indicate that the edge waviness is prevalent because of the tailing and imperfect dots of the engraved pattern, respectively. Figure 7a–e show: (a) an image of the printed circle pattern, (b) extracted edges of the pattern, (c) the measured radius with a designed radius and mean of measured radius (d) the widening, and (e) the histogram showing the waviness of the pattern.

The number of data points used to obtain the widening and edge waviness in the x-axis and y-axis (as shown in Figure 7d) was 1356. The pattern was widened by approximately 0.005 mm (0.59% of the pattern radius). The average, standard deviation, and a maximum deviation of the pattern radius were found to be 0.0099, 0.0067, and 0.0322 mm, respectively, owing to the edge waviness.

Table 2 shows the performance of the proposed algorithm compared to previous studies that evaluated the geometry and quality of printed patterns. The results show that the geometry and quality of the shape patterns can be clearly and reliably quantified using the proposed algorithm.

The proposed algorithm can be applied to complex patterns fabricated via additive manufacturing, such as those shown in Figure 3c,d in [19] and Figure 3a in [31], along with the coating and printing techniques used in this study, by combining the measurement methods for rectangle and circle patterns introduced in Section 2.1 and Section 2.2. In addition, the algorithm can be partially applied to irregular patterns, such as Figure 4b,c in [32], by aligning the printed pattern with the corresponding design pattern and comparing the differences between the pixels comprising the printed and corresponding design patterns.

## 4. Conclusions

This study proposed an advanced algorithm to facilitate reliable quantification of the geometry, widening, and edge waviness of printed patterns using image processing. Through the utilization of the three core concepts of the proposed algorithm, statistical measures were derived to quantitatively assess the geometry, widening, and edge waviness of the patterns based on all pixels constituting the boundary of a target pattern. Using the algorithm, we found that the rectangle and circle patterns in our study exhibited an average widening of 3.55% with a maximum deviation of 1.58%, based on an average of 1662 data points. The results showed that the proposed algorithm effectively and reliably evaluated the geometry and quality of the printed patterns. This indicates that the algorithm has potential applications in real-time classification, pattern quality evaluation, process optimization using statistical and AI-based approaches, as well as the foundation of IEC standards for shape pattern dimensions and printability.

## Figures and Tables

**Figure 1 nanomaterials-13-01597-f001:**
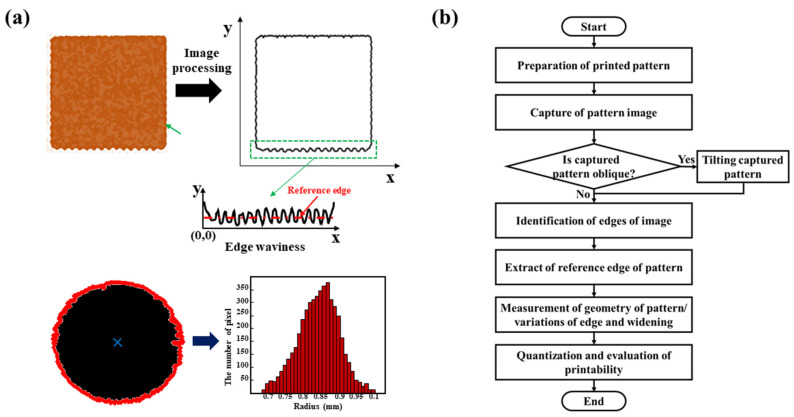
(**a**) Proposed quantification method and (**b**) its flowchart. The method incorporates three core ideas: (1) extraction of the edges of printed patterns and identification of the positions of pixels constituting the edges; (2) identification of the reference edges (reference sides and circles for rectangular and circular patterns, respectively), which are the best-fitting of the shape pattern; and (3) calculation of the different pixel positions of the edges related to the reference sides or circles.

**Figure 2 nanomaterials-13-01597-f002:**
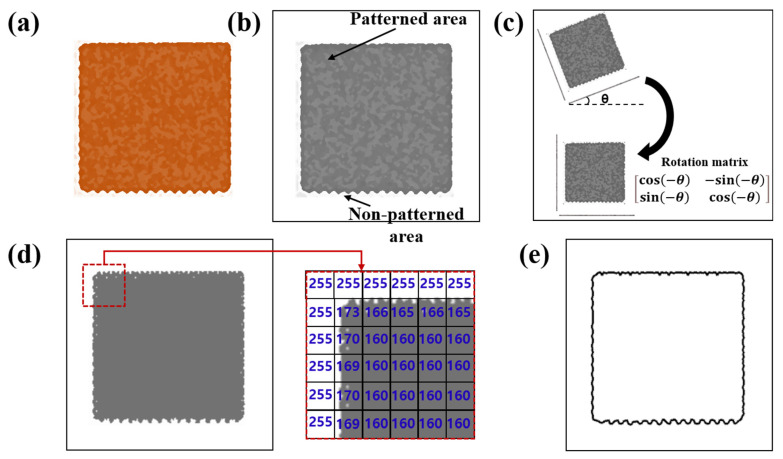
Process for quantifying the geometry and printability of a pattern using image processing. (**a**) Preparation of the printed pattern to be measured, (**b**) capture of an image of the pattern, (**c**) tilting of the pattern image, (**d**) distinguishing the pattern sides, and (**e**) identifying the reference edge of the printed pattern.

**Figure 3 nanomaterials-13-01597-f003:**
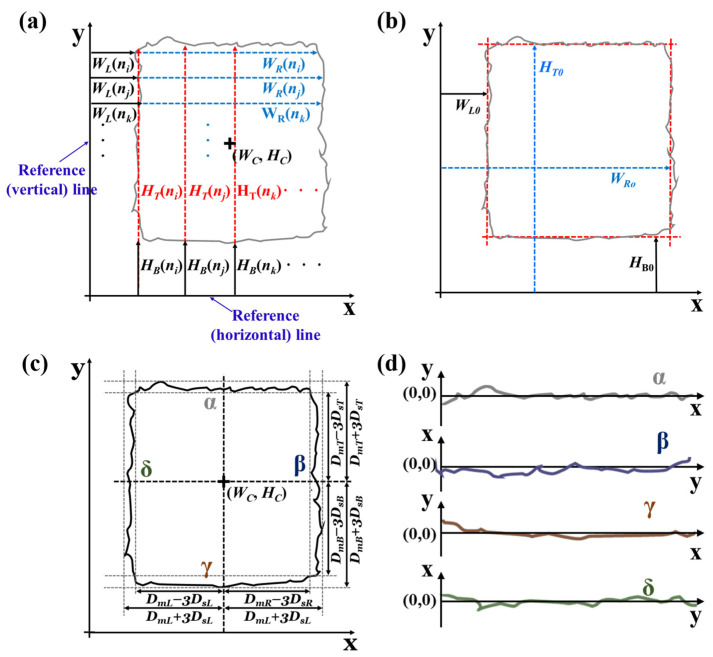
Schematic for the measurement of the geometry of a rectangle from the extracted pattern edges. First, vertical and horizontal reference lines that are parallel to the sides of the rectangle are drawn (**a**). The reference sides *W_L_*_0_, *W_R_*_0_, *H_B_*_0_, and *H_T_*_0_ were obtained (**b**). The data near the corners of the rectangle is excluded (**c**) and the edge waviness for each edge is obtained (**d**); In (**c**,**d**), α, β, γ, and δ are the top, right, bottom, and left sides of the target rectangle pattern, respectively.

**Figure 4 nanomaterials-13-01597-f004:**
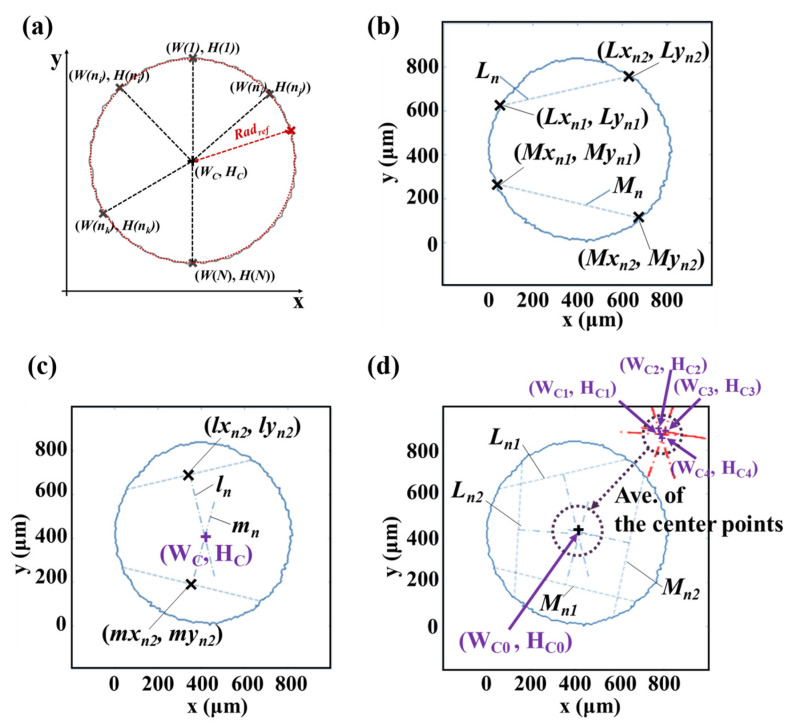
Schematic for measuring the geometry of a circle from the extracted pattern boundary. (**a**) shows a schematic of the radius of the circle. It is necessary to determine the center of the circle before measuring its radius. (**b**–**d**) illustrates how to identify the center of a circular pattern using lines passing through the edge.

**Figure 5 nanomaterials-13-01597-f005:**
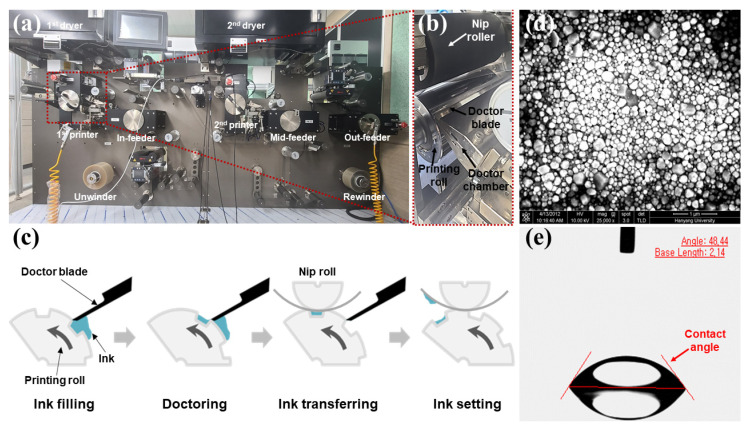
(**a**) Roll-to-roll machine and (**b**) direct gravure printing process used in this study. (**c**) Ink transferring phase in gravure printing (adapted from [24] with permission), (**d**) scanning electron microscopy (SEM) image, and (**e**) the contact angle of the ink solvent. In (**a**), the roll-to-roll machine comprises an unwinder, in-feeder, first and second printer, first and second dryer, out-feeder, and rewinder. In (**d**,**e**), the average particle size of the ink is 150 nm and the contact angle is 48.44°.

**Figure 6 nanomaterials-13-01597-f006:**
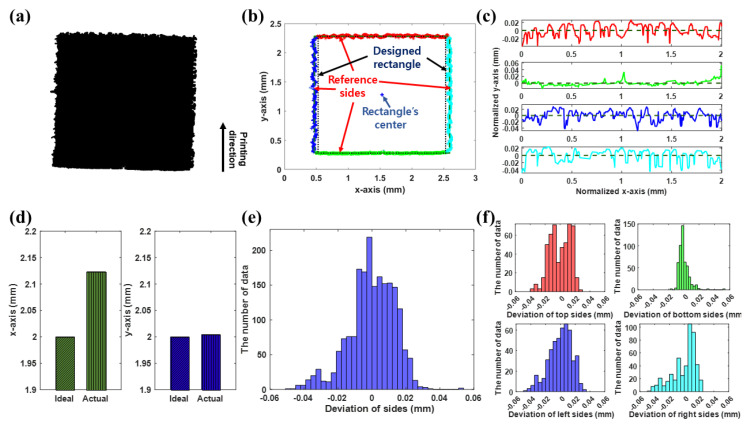
(**a**) Printed pattern image, (**b**,**c**) extracted edges of the pattern, (**d**) widening, and (**e**,**f**) histogram showing the waviness of the pattern (**e**) and each side (**f**). In (**b**,**c**,**f**), the red, green, blue, and cyan bold lines represent the top, bottom, left, and right edges of the pattern, respectively. Further, the dotted and dashed lines in (**b**,**c**) represent the designed pattern and reference sides, respectively. The number of data points used to obtain the widening and edge waviness in the x-axis and y-axis was 1968. As evident, the average widening and maximum deviation of the pattern image were 6.5% and 2.57% of pattern width, respectively.

**Figure 7 nanomaterials-13-01597-f007:**
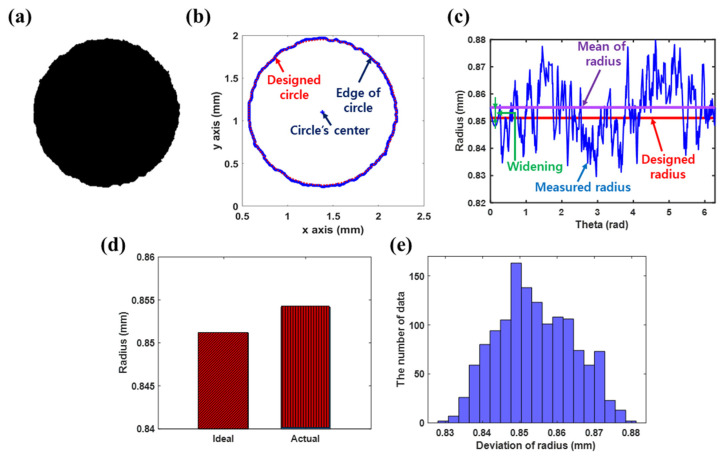
(**a**) Image of the printed circle pattern, (**b**) extracted edges of the pattern, (**c**) measured radius with a designed radius and mean of measured radius, (**d**) widening, and (**e**) histogram showing the waviness of the pattern. The number of data points used to obtain the widening and edge waviness in the x-axis and y-axis was 1356. As evident, the average widening and maximum deviation of the pattern image are 0.5% and 1.89% of the pattern radius, respectively.

**Table 1 nanomaterials-13-01597-t001:** Experimental conditions.

Conditions		Value
Ink property	Surface tension of ink	37 mN/m
	Ink viscosity	50 mN/m
	Average particle size	150 nm
Web property	Surface energy	10,000 cP
Printing conditions	Printing speed	5 m/min
	Tension	30 N
	Nip pressure	0.5 MPa
	Doctor pressure	0.5 MPa

**Table 2 nanomaterials-13-01597-t002:** Performance of the proposed algorithm compared to previous research.

Author	Pattern	Parameter for Printing Quality	Number of Measurement Data per Unit Sample	Ref.
Kapur, N	Strip (coating)	Fractional pickout	-	[11]
Kitsomboonloha, R. et al.	Line (printing)	Printed volume fraction (Avg. ^1^, Min. ^2^, Max. ^3^)	6	[12]
Nguyen, H.A.D. et al.	Line (printing)	Ink transfer ratio (ITR_A_)	-	[16]
		Pattern width (Avg., Min., Max.)		[16,17]
Hyun, W.J. et al.	Line (printing)	Pattern width (Avg., Min., Max.)	-	[18]
J. Park et al.	Line (printing)	Pattern width (Avg., Min., Max.)	18	[19]
		Widening ratio		
J. Noh et al.	Line (printing)	Pattern width (Avg.)	-	[20]
		Edge waviness (Avg.)		
J. Lee et al.	Line (printing)	Pattern width (Avg.)	9	[22]
		Local smudging (Avg.)		
		Continuity (Avg.)		
Nam, K.S. et al.	Line (printing)	Aspect ratio Pattern width (Avg., Min., Max.)	-	[25]
Lee, S.Y. et al.	Line (printing)	Edge waviness (Avg.)	6	[26]
Jeon, S.W. et al.	Line and mesh (printing)	Pattern width (Avg., Max.) Edge waviness (Avg., Max.)	30	[27]
Lee, M. et al.	T-shape (printing)	Pattern width (Avg., Min., Max.)	-	[28]
This work	Shapes (printing)	Pattern width, height and radius (Avg. and SD. ^4^)	Avg. 1662 ^5^ (Figure 6 and Figure 7)	-
		Widening ratio		
		Edge waviness (Avg. and SD)		

^1^ Avg.: Average; ^2^ Min.: Minimum; ^3^ Max.: Maximum; ^4^ SD.: Standard deviation. ^5^ The number of measurement data depends on the number of pixels in the pattern boundary.

## Data Availability

Not applicable.

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
