# Peer review of "Advanced Algorithm for Reliable Quantification of the Geometry and Printability of Printed Patterns"

_nanomaterials, 2023, doi:10.3390/nano13101597_

Round 1
Reviewer 1 Report
The manuscript develops an advanced algorithm to measure the geometry and printability of shape patterns in nanoparticle-based printed electronic devices. The algorithm involves extracting edges of printed patterns, identifying reference edges, and calculating different pixel positions of edges related to reference edges, enabling measurement of pattern geometry and printability including edge waviness and widening. Experimental verification has demonstrated the algorithm's reliability and efficacy in quantifying the geometry and quality of printed patterns, indicating potential applications in real-time pattern quality evaluation and process optimization. I would recommend this manuscript to be accepted for publication in Nanomaterials. Minor revisions are suggested below.
1. In the second step of quantifying geometry and printability of a pattern, does the determination of the image's tilt angle θ rely on the manual measurement?
2. For measuring the geometry of a rectangle from the extracted pattern edges, the reference lines are first drawn by measurer. Does this mean that the selection of sample points also relies on measurer?
3. For multiple sample points on a circle, there may be a large number of possible circle centers. Have the authors utilized any efficient algorithm to promptly calculate the mean value of the circle center’s coordinates?
4. As shown in Figure 6(a), there are significant differences in the waviness of the rectangular edges. It is suggested to evaluate the waviness of each edge separately and assess printing conditions accordingly.
5. The evaluation methods for the quality of the printed rectangular and circular patterns are developed in this manuscript. It would be desirable to supplement this algorithm's feasibility for application to complex geometrics or irregular patterns, such as Adv. Mater., 33, 2006361, (2021). In addition, the pattern quality evaluation algorithm proposed in the article should be applied to various advanced printing technologies, such as Mater. Today Nano, 16, 100136, (2021). It will make the paper attracting for a broad of audience.
good
Author Response
Thank you very much for your valuable comments, which were very useful to improve the paper. We tried to satisfy the your comments carefully. The answers to the comments are provided as separated file. In the revised manuscript, the revised contents are marked in yellow highlight as well.

Reviewer 2 Report
The manuscript talks about the advanced algorithm to reliably quantify the geometry and printed pattern. Below are my comments.
Remove fullstop in title.
Mention the percentage error in the abstract section.
Introduction part is very few. Have more research study and elaborate the Introduction section.
No any decision tree is involved in the flowchart?
Table 2 performance parameter is very few. Add more.
Its fine. Few grammatical sentences need the correction.
Author Response

(The authors gave the same response as above.)
